# Tree in Tree: from Decision Trees to Decision Graphs

**Bingzhao Zhu**
EPFL
Lausanne, Switzerland
Cornell University
Ithaca, NY, USA
bz323@cornell.edu

**Mahsa Shoaran**
EPFL
Lausanne, Switzerland
mahsa.shoaran@epfl.ch

## Abstract

Decision trees have been widely used as classifiers in many machine learning applications thanks to their lightweight and interpretable decision process. This paper introduces Tree in Tree decision graph (TnT), a framework that extends the conventional decision tree to a more generic and powerful directed acyclic graph. TnT constructs decision graphs by recursively growing decision trees inside the internal or leaf nodes instead of greedy training. The time complexity of TnT is linear to the number of nodes in the graph, and it can construct decision graphs on large datasets. Compared to decision trees, we show that TnT achieves better classification performance with reduced model size, both as a stand-alone classifier and as a base estimator in bagging/AdaBoost ensembles. Our proposed model is a novel, more efficient, and accurate alternative to the widely-used decision trees.

## 1 Introduction

Decision trees (DTs) and tree ensembles are widely used in practice, particularly for applications that require few parameters [1–5], fast inference [6–8], and good interpretability [9, 10]. In a DT, the internal and leaf nodes are organized in a binary structure, with internal nodes defining the routing function and leaf nodes predicting the class label. Although DTs are easy to train by recursively splitting leaf nodes, the tree structure can be suboptimal for the following reasons: (1) DTs can grow exponentially large as the depth of the tree increases. Yet, the root-leaf path can be short even for large DTs, limiting the predictive power. (2) In a DT, the nodes are not shared across different paths, reducing the efficiency of the model.

Decision trees are similar to neural networks (NNs) in that both models are composed of basic units. A possible way to enhance the performance of DTs or NNs is to replace the basic units with more powerful models. For instance, "Network in Network" builds micro NNs with complex structures within local receptive fields to achieve state-of-the-art performances on image recognition tasks [11]. As for DTs, previous work replaced the axis-aligned splits with logistic regression or linear support vector machines to construct oblique trees [1, 3, 6, 12–14]. The work in [5] further incorporates convolution operations into DTs for improved performance on image recognition tasks, while [1] replaces the leaf predictors with linear regression to improve the regression performance. Unlike the greedy training algorithms used for axis-aligned trees (e.g., Classification and Regression Trees or CART [15]), oblique trees are generally trained by gradient-based [3, 13, 14] or alternating [1, 6] optimization algorithms.

Inspired by the concepts of Network in Network [11] and oblique trees [6, 12], we propose a novel model, Tree in Tree (TnT), to recursively replace the internal and leaf nodes with micro decision trees. In contrast to a conventional tree structure, the nodes in a TnT form a Directed Acyclic Graph (DAG) to address the aforementioned limitations and construct a more efficient model. Unlike previous oblique trees that were optimized on a predefined tree structure [1, 5], TnT can learn graph

35th Conference on Neural Information Processing Systems (NeurIPS 2021).

connections from scratch. The major contributions of this work are as follows: *(1) We extend decision trees to decision graphs and propose a scalable algorithm to construct large decision graphs. (2) We show that the proposed algorithm outperforms existing decision trees/graphs, either as a stand-alone classifier or base estimator in an ensemble, under the same model complexity constraints. (3) Rather than relying on a predefined graph/tree structure, the proposed algorithm is capable of learning graph connections from scratch (i.e., starting from a single leaf node) and offers a fully interpretable decision process.* We provide a Python implementation of the proposed TnT decision graph at https://github.com/BingzhaoZhu/TnTDecisionGraph.

## 2   Related work

Decision graph (DG) is a generalization of the conventional decision tree algorithm, extending the tree structure to a directed acyclic graph [16, 17]. Despite similarity in using a sequential inference scheme, training and optimizing DGs is more challenging due to the large search space for the graph structure. The work in [16] and [18] proposed a greedy algorithm to train DGs by tentatively joining pairs of leaf nodes at each training step (NDG, Algorithm 1). [19] constructed

---

**Algorithm 1:** Naive decision graph (NDG) [16]

1   $G \leftarrow$ initialize graph with a leaf node;
2   **for** $i \leftarrow 1$ **to** $N$ **do**
3     **for** *each leaf node* $(l_i) \in G$ **do**
4       Find the maximum gain $(g_i)$ if we split $l_i$ ;
5     **for** *each pair of leaf nodes* $(l_i, l_j) \in G$ **do**
6       Record gain $(g_{i,j})$ if we merge $l_i$ and $l_j$;
7     Split/merge nodes to maximize gain;
8   Note: The split operation has a model complexity penalty ($C$) for creating an internal node.

---

the DG as a Markov decision process, where base estimators were cascaded in the form of a data-dependent AdaBoost ensemble. [20] combined multiple binary classifiers to construct a DAG for efficient multi-class classification. While [19] and [20] constructed DAGs with specific settings (e.g., Adaboost [19] and multi-class classification [20]), these methods do not provide a general extension of decision trees. [21] improved NDG by jointly optimizing split nodes and their connections to next-layer nodes. However, simultaneously learning the split and branching structure has no exact solution and relies on search-based algorithms to reach local minimum. Alternatively, in this work, we revisit the concept of decision graphs by exploiting recent advances in non-greedy tree optimization algorithms [6, 12, 14, 22]. Our proposed Tree in Tree algorithm can construct DGs as a more accurate and efficient alternative to the widely-used decision trees, both as stand-alone classifiers and as weak learners in the ensembles.

Conventional decision tree learning algorithms such as CART [15] and its variations follow a greedy top-down growing scheme. Recent work has focused on optimizing the structure of the tree [22–24]. However, constructing an optimal binary DT is NP-hard [25] and optimal trees are not scalable to large datasets with many samples and features [22–24]. Recent studies have further developed scalable algorithms for non-greedy decision tree optimization, with no guarantee on tree optimality [1, 3, 6, 12–14]. Such scalable approaches can be categorized into two groups: tree alternating optimization (TAO) [1, 6] and gradient-based optimization [3, 12–14].

TAO decomposes the tree optimization problem into a set of reduced problems imposed at the node levels. The work in [6] applied the alternating optimization to both axis-aligned trees and sparse oblique trees. Later, [1] extended TAO to regression tasks and ensemble methods. Unlike TAO, gradient-based optimization requires a differentiable objective function, which can be obtained by different methods. For example, [12] derived a convex-concave upper bound of the empirical loss. [13] and [3] considered a soft (i.e., probabilistic) split at the internal nodes and formulated a global objective function. The activation function for soft splits was refined in [14] to enable conditional inference and parameter update. Both TAO and gradient-based optimization operate on a predefined tree structure and optimize the parameters of the internal nodes.

The proposed Tree in Tree algorithm aims to optimize the graph/tree structure by growing micro decision trees inside current nodes. Compared to the greedy top-down tree induction [15], Tree in Tree solves a reduced optimization problem at each node, which is enabled via non-greedy tree alternating optimization techniques [6]. Compared to NDG, TnT employs a non-greedy process to construct decision graphs, which leads to an improved classification performance (discussed in later sections). Compared to axis-aligned decision trees (e.g., TAO [1, 6], CART [15]), TnT extends the

tree structure to a more accurate and compact directed acyclic graph, in which nodes are shared across multiple paths.

# 3 Methods

In this work, we consider a classification task with input and output spaces denoted by $\mathcal{X} \subset \mathbb{R}^D$ and $\mathcal{Y} = \{1, ..., K\}$, respectively. Similar to conventional decision trees, a decision graph classifier $G$ consists of internal nodes and leaf nodes. Each internal node is assigned a binary split function $s(\cdot; \theta) : \mathcal{X} \to [left\_child, right\_child]$ parametrized by $\theta$, which defines the routing function of a graph. For axis-aligned splits, $\theta$ indicates a feature index and a threshold. The terminal nodes (with no children) are named leaf nodes and indicate the class labels.

## 3.1 Decision graph

As an extension to the tree structure, decision graphs organize the nodes into a more generic directed acyclic graph. In this work, we limit our discussion to axis-aligned binary DTs/DGs in which each internal node compares a feature value to a threshold to select one of the two child nodes. Similar to the sequential inference process in DTs, the test samples in a DG start from the root and successively select a path at the internal nodes until a leaf node is reached. The main differences between binary DTs and DGs are the following: (1) In DTs, each child node has one parent node. However, DGs allow multiple parent nodes to share the same child node. Therefore, DG can combine the nodes with similar behaviors (e.g., similar split functions) to reduce model complexity. (2) In binary DTs, the number of leaf nodes is always greater than the internal nodes by one. In DGs, however, $\#Leaves \leq \#Internals + 1$, since multiple internal nodes can share the same leaf node. Furthermore, there exists a unique path to reach each leaf node in a tree structure, which does not hold within DGs. (3) The model complexity of a DT is often quantified by the number of internal or leaf nodes. However, we can post-process a DG by merging the leaf nodes with the same class label. As a result, DGs have a minimum leaf node count equal to the number of classes. Therefore, we use the number of splits (i.e., internal nodes) to quantify the model complexity of a DG.

## 3.2 Tree in Tree

We propose a novel algorithm named Tree in Tree as a scalable method to construct large decision graphs. Conventional DT training algorithms (e.g., CART) are greedy and recursively split the leaf nodes to grow a deep structure, without optimizing the previously learned split functions. *The key difference between the proposed TnT model and conventional approaches lies in the optimization of the internal nodes. TnT fits new decision trees in place of the internal/leaf nodes and employs such micro DTs to construct a directed acyclic graph.* Overall, the proposed TnT model is a novel extension to the conventional decision trees and generates accurate predictions by routing samples through a directed acyclic graph.

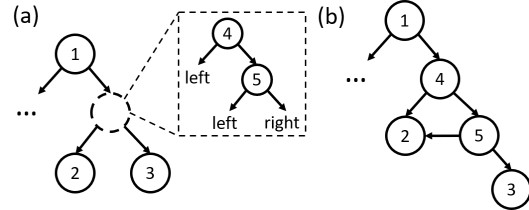

Figure 1: (a) The growing phase of TnT. The micro decision tree (in dashed box) replaces an internal node (dashed circle). Compared to a single node, the substitute micro tree can provide a more powerful split function. (b) The merging phase of TnT. We merge the fitted micro tree into the current structure to create a directed acyclic graph.

Figure 1 shows the high-level procedure for training a decision graph with the proposed TnT algorithm. Assuming a starting decision graph (e.g., a decision tree or a single leaf node), our goal is to grow a larger model with improved predictive power. In the growing phase of TnT (Fig. 1(a)), we replace a node (dashed circle) with a micro decision tree with multiple splits to enable more accurate decision boundaries. In the merging phase (Fig. 1(b)), the micro decision tree is merged into the starting model to construct a TnT decision graph, in which a child node (node 2) may have multiple parent nodes (node 4 and 5).

**Growing the graph from internal nodes** We consider the training of a decision graph as an optimization problem with the aim of minimizing the loss function on the training data:

$$\min \sum_{x, y \in \mathbb{X}, \mathbb{Y}} L\left(y, G(x; \Theta)\right). \tag{1}$$

TnT grows the decision graph $G(\cdot; \Theta)$ from an arbitrary internal node $n_i \in G$ with the split function $s(\cdot; \theta_i)$. $\theta_i$ denotes the trainable parameters of $n_i$ including a feature index and a threshold for axis-aligned splits. The overall goal is to replace $n_i$ with a decision tree $t_i$ and minimize the loss function as indicated in (1). All other nodes remain unchanged as we train $t_i$.

Let us consider a subset of samples ($x_{subset} \in \mathbb{X}_{subset}, y_{subset} \in \mathbb{Y}_{subset}$) that is sensitive to the split function $s(\cdot; \theta_i)$, as defined by the following expression:

$$G_{n_i \to left}(x_{subset}; \Theta \backslash \theta_i) \neq G_{n_i \to right}(x_{subset}; \Theta \backslash \theta_i), \tag{2}$$

where $\Theta \backslash \theta_i$ denotes the parameters of all nodes in $G$ excluding $n_i$. Growing the graph from $n_i$ does not change $\Theta \backslash \theta_i$ since all other nodes are fixed as we solve the reduced optimization problem at $n_i$. $G_{n_i \to left}$ sends the samples to the left child at $n_i$ (i.e., $s(\cdot; \theta_i) \to left\_child$) while $G_{n_i \to right}$ routes the samples to the right child at $n_i$. With $\Theta \backslash \theta_i$ being fixed, the output of decision graph only depends on $\theta_i$ (i.e., $s(\cdot; \theta_i)$)

$$G(x; \Theta) = \begin{cases} G_{n_i \to left}(x; \Theta \backslash \theta_i) & \text{if } s(x; \theta_i) \to left\_child \\ G_{n_i \to right}(x; \Theta \backslash \theta_i) & \text{if } s(x; \theta_i) \to right\_child. \end{cases} \tag{3}$$

Since $L(y, G_{n_i \to left}(x; \Theta \backslash \theta_i)) \neq L(y, G_{n_i \to right}(x; \Theta \backslash \theta_i))$ only if the inequality (2) holds, we can solve the reduced optimization problem at node $n_i$ based on the subset ($\mathbb{X}_{subset}, \mathbb{Y}_{subset}$) instead of using the entire training set:

$$\min_{\theta_i} \sum_{\substack{x \in \mathbb{X}_{subset} \\ y \in \mathbb{Y}_{subset}}} L(y, G(x; \Theta)). \tag{4}$$

Having Equation (3), the optimization problem (4) has a closed-form solution as follows:

$$t_i^*(x) := \begin{cases} left\_child & \text{if } L(y, G_{n_i \to left}(x; \Theta \backslash \theta_i)) < L(y, G_{n_i \to right}(x; \Theta \backslash \theta_i)) \\ right\_child & \text{if } L(y, G_{n_i \to right}(x; \Theta \backslash \theta_i)) < L(y, G_{n_i \to left}(x; \Theta \backslash \theta_i)). \end{cases} \tag{5}$$

Equation (5) defines the optimal split function at the internal node $n_i$ which is used to fit the micro decision tree $t_i$. With other nodes being fixed, we show that the overall loss function of $G$ can be minimized by pursuing the optimal spilt function at an arbitrary internal node $n_i$. Rather than using a simple axis-aligned split, the proposed TnT algorithm learns a complexity-constrained decision tree to better approximate the optimal split function (Equation (5)).

**Growing the graph from leaf nodes**    Growing from the leaf nodes is a standard practice in greedy training algorithms, where we recursively split the leaf nodes to achieve a deeper tree with a better fit on the training data [15]. In TnT, we replace the leaf predictors with decision trees. Let $G(\cdot; \Theta)$ be a decision graph and $n_l \in G$ an arbitrary leaf node with a constant class label $l(\cdot; \theta_l) = c$. Our goal is to minimize the overall loss function $L(\mathbb{Y}, G(\mathbb{X}; \Theta))$ by replacing the leaf predictor $l(\cdot; \theta_l)$ with a micro decision tree $t_l(x)$.

Consider the subset of samples ($\mathbb{X}_{subset}, \mathbb{Y}_{subset}$) that visit the leaf node $n_l$. Minimization of the loss function (1) can be expressed as

$$\min_{\theta_l} \sum_{\substack{x \in \mathbb{X}_{subset} \\ y \in \mathbb{Y}_{subset}}} L(y, t_l(x)), \tag{6}$$

where the minimum is simply achieved at $t_l^*(x) := y$ for $x, y \in \mathbb{X}_{subset}, \mathbb{Y}_{subset}$ (i.e., the ideal leaf predictor). We build a decision tree to approximate the ideal leaf predictor.

### 3.3   Learning procedure

Unlike the learning procedures in [1, 6] which require a predefined tree structure, our proposed TnT algorithm grows a decision graph from a single leaf node. The training of TnT decision graphs is an iterative process that follows a *grow-merge-grow- · · · · -merge* alternation. Algorithm 2 shows the pseudocode to train a TnT decision graph. Lines 9-14 find the subset of data samples $\mathcal{X}_{subset}, \mathcal{Y}_{subset}$ that is sensitive to the internal split functions or leaf predictors at each node, and grow micro decision trees. In the internal nodes, $\mathcal{Y}_{subset}$ represents binary labels for the left or right child (i.e., not the label of the training set). Line 17 grows micro decision trees according to the growing phase of the TnT. Line 18 merges the trees into the graph structure. In Algorithm 2, $N_1$ is the number of merging phases that micro trees are merged into the graph. $N_2$ is the number of rounds to grow and optimize micro trees, similar to the number of iterations in the tree alternating optimization algorithm [1, 6].

**Regularization** Regularization is critical to limit model complexity and prevent overfitting of a decision tree and it is similarly required for TnT decision graphs. In the growing phase of a TnT (either from internal or leaf nodes), the subsets of samples $\mathcal{X}_{subset}, \mathcal{Y}_{subset}$ at different nodes may have various sizes. Therefore, we need a robust regularization technique to operate across all nodes of the TnT and to train the micro decision trees without overfitting on small subsets. In this work, we propose to use the sample-weighted cost complexity pruning approach [26, 27]. We prune micro decision trees by minimizing $R(t_i) + C_i|t_i|$, where $R(t_i)$ is the misclassification measurement and $|t_i|$ denotes the tree complexity. We calculate $R(t_i)$ using Gini impurity and measure $|t_i|$ by counting the number of splits [15]. $C_i$ is the sample-weighted regularization coefficient calculated by

$$C_i = C \frac{\#\mathcal{X}}{\#\mathcal{X}_{subset,i}}, \tag{7}$$

where $\#\mathcal{X}_{subset,i}$ is the sample count of subset at node $n_i$. $C$ is a hyperparameter of the TnT and is used to control the pruning strength and tune the model complexity (# splits). For a smaller subset, we need to apply a stronger cost complexity pruning to prevent overfitting.

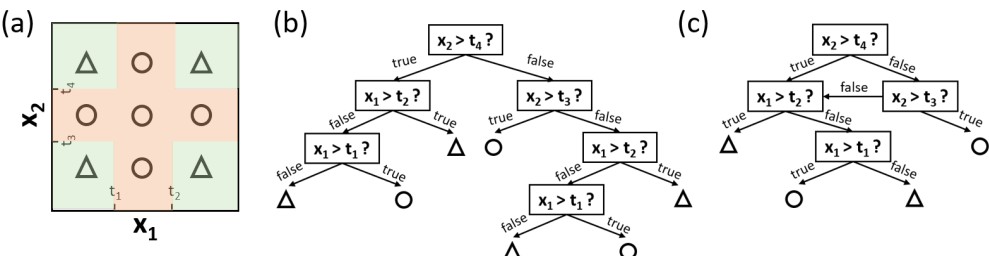

Figure 2: Comparison of DT and TnT decision graph on synthetic data; (a) A toy classification task with desired axis-aligned boundaries. $x_1, x_2$ and $t_1 - t_4$ denote two features and four thresholds, respectively. Different markers represent binary class labels. (b) A decision tree requires at least six splits to classify the data. (c) A TnT decision graph only requires four binary splits on the same task.

**Fine-tune and post pruning** The TnT decision graphs are compatible with Tree Alternating Optimization (TAO [6]), previously proposed to optimize decision trees. We used TAO to fine-tune the TnT decision graphs, which led to slight improvements in classification accuracy. The pseudocode for TnT fine-tune algorithm is provided in the supplementary materials. A post pruning process is further applied to TnT decision graphs to remove the dead nodes. A node is pruned if no training

---

**Algorithm 2:** Tree in Tree (TnT)

**Data:** Training set $\mathcal{X}, \mathcal{Y}$
**Result:** TnT decision graph $G$ fitted on the training set

1   $\{infer(n, \mathcal{X})$ denotes the forward inference of data $\mathcal{X}$ starting from node $n\}$;
2   {Nodes are visited in the breadth-first order};
3   $G \leftarrow$ initialize graph with a leaf node;
4   **for** $i_1 \leftarrow 1$ **to** $N_1$ **do**
5     **for** $i_2 \leftarrow 1$ **to** $N_2$ **do**
6       **for** *each node* $(n_i) \in G$ **do**
7         Samples that visit $n_i$: $\mathcal{X}_i, \mathcal{Y}_i \subset \mathcal{X}, \mathcal{Y}$;
8         **if** $n_i$ *is an internal node* **then**
9           $\mathcal{Y}_{i,left} \leftarrow infer(n_i.left\_child, \mathcal{X}_i)$;
10           $\mathcal{Y}_{i,right} \leftarrow infer(n_i.right\_child, \mathcal{X}_i)$;
11           $index\_left \leftarrow (\mathcal{Y}_i = \mathcal{Y}_{i,left}$ **and** $\mathcal{Y}_i \neq \mathcal{Y}_{i,right})$ ;
12           $index\_right \leftarrow (\mathcal{Y}_i = \mathcal{Y}_{i,right}$ **and** $\mathcal{Y}_i \neq \mathcal{Y}_{i,left})$ ;
13           $\mathcal{X}_{subset}, \mathcal{Y}_{subset} \leftarrow$ copy samples from $\mathcal{X}_i, \mathcal{Y}_i$ at $(index\_left$ **or** $index\_right)$;
14           $\mathcal{Y}_{subset}[index\_left] \leftarrow 0, \mathcal{Y}_{subset}[index\_right] \leftarrow 1$;
15         **else if** $n_i$ *is a leaf node* **then**
16           $\mathcal{X}_{subset} \leftarrow \mathcal{X}_i, \mathcal{Y}_{subset} \leftarrow \mathcal{Y}_i$;
17         Grow a micro tree $t_i$ on subset $\mathcal{X}_{subset}, \mathcal{Y}_{subset}$ in place of $n_i$;
18     Merge $t_i$ into the current decision graph $G$ for all nodes $(n_i \in G)$

samples travel through that node. Post pruning can result in a more compact decision graph and reduce the number of splits without affecting the training accuracy.

**Time complexity**   Compared to decision trees, decision graphs offer an enriched model structure, which increases the complexity of learning the graph structure. Previous work constructed decision graphs by tentatively merging two leaf nodes at each training step, with a time complexity of $O(N_l^2)$, where $N_l$ is the number of leaf nodes [16]. Since the proposed TnT algorithm generates new splits by growing micro decision trees inside the nodes, the dataset is initially sorted in $O(mklog(m))$ for $m$ samples and $k$ features. The time complexity for creating a new split depends on the dataset (i.e., $O(mk)$) and not on the size of the graph. As the graph grows larger, the TnT algorithm optimizes each node for $N_1 * N_2$ times in the worst case (Algorithm 2). Since $N_1$ and $N_2$ are hyperparameters that were fixed in this work ($N_1 = 2, N_2 = 5$, the choice of $N_1$ and $N_2$ will be discussed in the following section), TnT exhibits a linear time complexity to the number of nodes, $O(nmk + mklog(m))$ with $n$ being the number of nodes. Testing our Python implementation on an Intel i7-9700 CPU, it took 325.3 seconds to build a TnT of 1k splits on the MNIST dataset (60k samples, 784 features, 10 classes).

**Synthetic data**   We first construct a synthetic classification dataset to show the potential benefits of TnT over conventional decision tree algorithms (e.g., CART). Figure. 2(a) visualizes the two-dimensional data distribution with one class on the corners and the other class elsewhere. To achieve optimal decision boundaries, a conventional decision tree requires six splits (Fig. 2(b)), whereas TnT only requires four splits to generate the same decision boundaries (Fig. 2(c)). *By sharing nodes among different decision paths in a graph, TnT enables a more compact model with fewer splits compared to a conventional DT.*

## 4   Experiments: TnT as a stand-alone classifier

We test the TnT decision graph as a stand-alone classifier and benchmark it against several state-of-the-art decision tree/graph algorithms with axis-aligned splits, including classification and regression trees (CART [15]), tree alternating optimization (TAO [6]), and the naive decision graph (NDG [16]). We did not include decision jungles [21] in our comparison since no implementation was provided by the authors. We also implemented the TnT algorithm in two different settings: with or without fine-tuning. We observed that the proposed TnT algorithm consistently achieves a superior performance under similar complexity constraints on multiple datasets.

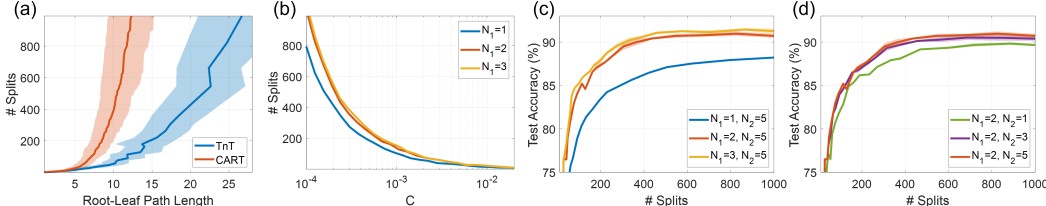

Figure 3: (a) The number of splits as a function of the root-leaf path length. The standard deviation across different samples is shown by shaded areas. (b) The number of splits vs. regularization coefficient $C$. (c, d) Test performance using different hyperparameter settings on the MNIST dataset. The default setting ($N_1 = 2, N_2 = 5$) is plotted in both figures for comparison.

In the worst-case scenario, the number of nodes increases exponentially with the depth of a tree, which prevents DTs from growing very deep. However, this limitation does not apply to TnT decision graphs. Figure 3(a) illustrates the average length of the root-leaf path as a function of model complexity for TnT and CART. With 1000 splits, the average decision depth of the best-first CART is 12.3, whereas the TnT decision graph has a mean depth of 27.3. In the best-first decision tree induction, we add the best split in each step to maximize the objective [28]. *Therefore, TnT can achieve a much "deeper" model without significantly increasing the number of splits.* The regularization coefficient $C$ is used to control the complexity of decision graphs in TnT. The number of splits decreases as we increase the pruning strength $C$ (Fig. 3(b)). Figures 3(c, d) compare the effect of different hyperparameter settings $(N_1, N_2)$. *We note that the proposed TnT decision graph is a superset of decision trees and that TnT can reduce to a DT learning algorithm under certain conditions.* With $N_1 = 1$, Algorithm 2 replaces

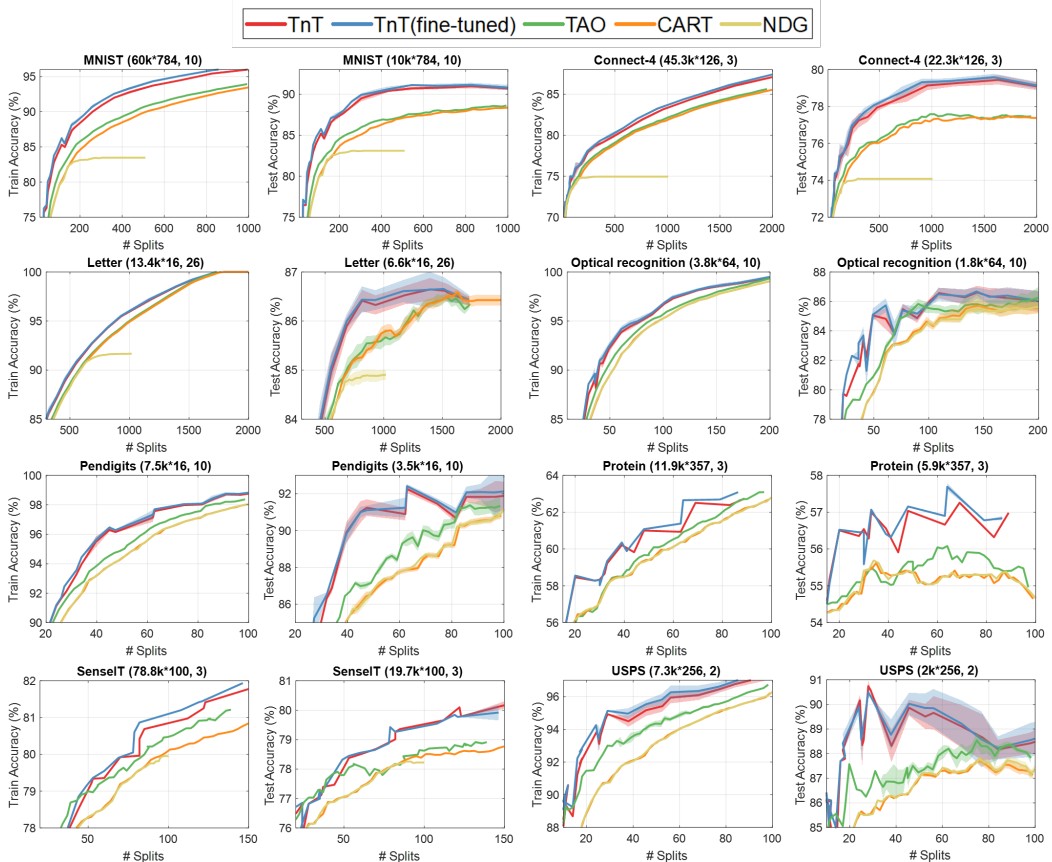

Figure 4: Model comparison in terms of train and test accuracy on multiple classification tasks. The following axis-aligned decision trees/graphs are included: **TnT (this work):** We implement the proposed TnT decision graph at various complexity levels. Hyperparameters are fixed at $N_1 = 2, N_2 = 5$ on all tasks. **TnT (fine-tuned):** The alternating optimization algorithm is used to fine-tune the TnT. **TAO:** The tree alternating optimization algorithm is applied to axis-aligned decision trees [29]. **CART:** Classification and regression trees trained in a best-first manner to assess the optimal tree structure under certain complexity constraint [15, 28]. **NDG:** The naive decision graph trained with Algorithm 1 [16]. The complexity penalty is fixed at $C = 3e - 4$ on all tasks. Dataset statistics are indicated on top of each figure with the following format (# Train/Test samples * # Features, # Classes).

a single leaf node with a decision tree, which is equivalent to training a CART with cost complexity pruning. In general, higher values of $N_1$ and $N_2$ can lead to a better classification performance. In the following experiments, we set the hyperparameters as $N_1 = 2, N_2 = 5$. A marginal improvement in classification performance can be obtained by increasing $N_1$ and $N_2$, at the cost of increased training time.

Figure 4 compares the proposed TnT decision graphs with axis-aligned decision trees/graphs previously reported. We include the following datasets: MNIST, Connect-4, Letter, Optical reconstruction, Pendigits, Protein, SenseIT, and USPS from the UCI machine learning repository [30] and LIBSVM Dataset [31] under Creative Commons Attribution-Share Alike 3.0 license. The statistics of datasets including the number of train/test instances, number of attributes, and number of classes are shown in Fig. 4. If a separate test set is not available for some tasks, we randomly partition 33% of the entire data as test set. For all models, we repeat the training procedure five times with different random seeds. The mean classification accuracy is plotted in Fig. 4 with shaded area indicating the standard deviation across trials. The proposed Tree in Tree (TnT) algorithm outperforms axis-aligned decision trees such as TAO [6, 29] and CART [15], as well as NDG which is also based on axis-aligned decision graphs [16]. We also present the results for TnT(fine-tuned), which employs alternating optimization to fine-tune the TnT and slightly improve the classification performance.

Table 1: Comparison of TnT and CART at optimal split count (#S, determined by cross-validation). Mean test accuracy (±standard deviation) are calculated on 5 independent trials.

| model | MNIST | | Connect-4 | | Letter | | Optical recognition | |
|---|---|---|---|---|---|---|---|---|
| | accuracy | #S | accuracy | #S | accuracy | #S | accuracy | #S |
| TnT | **90.87±0.31** | **600** | **78.85±0.46** | **864** | **86.62±0.02** | **1.2k** | **86.32±0.24** | **174** |
| CART | 88.59±0.14 | 1.1k | 77.23±0.01 | 931 | 86.26±0.15 | 1.3k | 85.56±0.46 | 193 |

| model | Pendigits | | Protein | | SenseIT | | USPS | |
|---|---|---|---|---|---|---|---|---|
| | accuracy | #S | accuracy | #S | accuracy | #S | accuracy | #S |
| TnT | **92.61±0.53** | **125** | **57.26** | **69** | **80.48±0.42** | **198** | **88.76±1.36** | **31** |
| CART | 91.74±0.13 | 166 | 55.30 | 76 | 79.40 | 345 | 87.35±0.15 | 109 |

The node-sharing mechanism in TnT effectively regularizes the growth of the graph and can increase test performance compared to greedy training of trees (e.g., CART). To show the reduction of variance in TnT decision graphs, we removed the complexity constraints on both models and selected the hyperparameters that achieve the highest cross-validation accuracy on the training set. Table 1 compares the proposed TnT method with greedy training of trees. Overall, TnT achieves a better test accuracy with reduced model complexity compared to CART.

**Visualization**   Similar to decision trees, TnT decision graphs enjoy a fully interpretable and visualizable decision process. Figures 5(a-c) visualize the TnT decision graphs with 20, 129, and 1046 splits, respectively. We use different node colors to indicate the dominant class labels. A node will have a dominant class if most samples at that node belong to the same class. We show the nodes in blue if class labels are mixed (i.e., no class label contributes to greater than 50% of the samples visiting that node). As the graph grows larger, TnT performs better on the MNIST dataset, achieving improved classification accuracy on both training and testing sets.

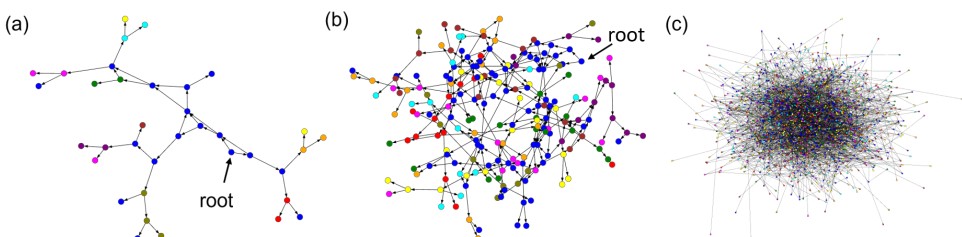

Figure 5: Visualization of TnT decision graphs at various complexity levels. (a) TnT with 20 internal nodes and 16 leaf nodes (train/test accuracy: 70.41%/71.75% on MNIST classification task). (b) 129 internals and 75 leaves (train/test accuracy: 85.54%/85.49%). (c) 1046 internals and 630 leaves (train/test accuracy: 96.04%/90.56%). Different node colors represent dominant class labels (more than 50% of samples belong to the same class). Nodes are shown in blue if no dominant class is found.

## 5   Experiments: TnT in the ensemble

Decision trees are widely used as base estimators in ensemble methods such as bagging and boosting. Random Forests apply a bagging technique to decision trees to reduce variance [32], in which each base estimator is trained using a randomly drawn subset of data with replacement [33]. As opposed to bagging, boosting is used as a bias reduction technique where base estimators are incrementally added to the ensemble to correct the previously misclassified samples. Popular implementations of the boosting methods include AdaBoost [34] and gradient boosting [35, 36]. Both AdaBoost and bagging use classifiers as base estimators, whereas the gradient boosting methods require regressors [35, 36]. Although we argue that the proposed TnT algorithm can be applied to regression tasks with a slight modification in the objectives, it is beyond the scope of this paper to demonstrate TnTs as regressors.

Here, we use the TnT decision graphs as base estimators in the bagging (TnT-bagging) and AdaBoost (TnT-AdaBoost) ensembles. *Our goal is to replace decision trees with the proposed TnT classifiers in*

*ensemble methods and compare the performance under various model complexity constraints.* The ensemble methods are implemented using the scikit-learn library in Python (under the 3-Clause BSD license) [37]. We change the ensemble complexity by tuning the number of base estimators (#E) and the total number of splits (i.e., internal nodes, #S). For Random Forest with 100 trees, we remove the limit on #S. Thus, the trees are allowed to grow as large as possible to better fit on training data. Note that TnT has additional hyperparameters such as $N_1$ and $N_2$. We set the hyperparameters as $N_1 = 2, N_2 = 5$ throughout the experiments so that the TnT and tree ensembles share a similar hyperparameter exploration space.

Table 2: Comparison of TnT-based ensembles with conventional random forest and AdaBoost. Mean train and test accuracy ($\pm$ standard deviation) are calculated across 5 independent trials. We tune the ensemble size (#E, the number of base estimators) and split count (#S) to change the complexity of the ensemble. Dataset statistics are given in the format: Dataset name (# Train/Test samples * # Features, # Classes). Six additional datasets are included in the supplementary materials.

| model | | #E | #S | train | test | | #E | #S | train | test |
|---|---|---|---|---|---|---|---|---|---|---|
| TnT-bagging | | 5 | 4.8k | **97.46±0.16** | **93.65±0.24** | | 5 | 4.6k | **84.42±0.19** | **80.61±0.18** |
| Random Forest | | 5 | 4.8k | 96.55±0.36 | 92.31±0.57 | | 5 | 4.6k | 83.60±0.12 | 79.21±0.19 |
| TnT-AdaBoost | | 5 | 640 | **90.26** | 88.38 | | 5 | 450 | **77.75±0.16** | **77.39±0.19** |
| AdaBoost | MNIST (60k/10k*784, 10) | 5 | 640 | 89.75 | **88.61** | Connect-4 (45.3k/22.3k*126, 3) | 5 | 450 | 77.28 | 76.74 |
| TnT-bagging | | 10 | 9.6k | **98.28±0.06** | **94.92±0.20** | | 10 | 9.2k | **85.11±0.05** | **81.44±0.14** |
| Random Forest | | 10 | 9.6k | 97.44±0.18 | 93.64±0.38 | | 10 | 9.2k | 84.21±0.12 | 79.85±0.20 |
| TnT-AdaBoost | | 10 | 1.4k | **95.09±0.09** | **92.36±0.13** | | 10 | 940 | **80.10±0.23** | **78.94±0.29** |
| AdaBoost | | 10 | 1.4k | 94.28 | 91.49 | | 10 | 940 | 79.69 | 78.37 |
| TnT-bagging | | 20 | 19.2k | **98.64±0.06** | **95.57±0.14** | | 20 | 18.3k | **85.66±0.12** | **81.93±0.13** |
| Random Forest | | 20 | 19.2k | 97.90±0.12 | 94.36±0.19 | | 20 | 18.3k | 84.57±0.08 | 80.39±0.09 |
| TnT-AdaBoost | | 20 | 2.9k | **98.03±0.11** | **94.49±0.21** | | 20 | 1.8k | 82.46±0.41 | 80.53±0.50 |
| AdaBoost | | 20 | 2.9k | 97.70 | 94.04 | | 20 | 1.8k | **82.77** | **81.14** |
| TnT-bagging | | 100 | 111k | 99.09±0.03 | **96.11±0.09** | | 100 | 143k | 88.44±0.07 | **82.84±0.02** |
| Random Forest | | 100 | 292k | **100** | 95.72±0.17 | | 100 | 718k | **100** | 82.33±0.10 |

Table 2 compares the performance of TnT ensembles with that of decision tree ensembles on two datasets. A complete comparison table on eight datasets is included in the supplementary materials. Since the bagging method can effectively reduce variance, we use large models (i.e., TnTs/decision trees with many splits) as the base estimator. On the contrary, TnTs/decision trees with few splits are used in the AdaBoost ensemble, given that boosting can decrease the bias error. According to Table 1, TnT-bagging is almost strictly better than Random Forest under the same model complexity constraints, indicating that TnT decision graphs outperform decision trees as base estimators. TnT-AdaBoost also outperforms AdaBoost in most cases, showing the advantage of TnT over decision trees. However, we observe a few exceptions in the TnT-AdaBoost vs. AdaBoost comparison, as weak learners with high bias (e.g., decision stumps) are also suitable for boosting ensembles. Overall, the TnT ensembles (TnT-bagging, TnT-AdaBoost) achieve a higher classification accuracy compared to decision trees when used in similar ensemble methods (Random Forest, AdaBoost).

## 6 Discussions

**Broader impact** Recently, the machine learning community has seen different variations of decision trees [1, 3, 5, 6, 12–14]. In this paper, we present the TnT decision graph as a more accurate and efficient alternative to the conventional axis-aligned decision tree. However, the core idea of TnT (i.e., growing micro trees inside nodes) is generic and compatible with many existing algorithms. For example, linear-combination (oblique) splits can be easily incorporated into the proposed TnT framework. Specifically, we can grow oblique decision trees inside the nodes to construct an oblique TnT decision graph (Line 17 of Algorithm 2). In addition to oblique TnTs, the proposed TnT framework is also compatible with regression tasks. As suggested in [1], we may grow decision tree regressors (rather than DT classifiers) inside the leaf nodes to construct TnT regressors, which remains as our future work. Overall, our results show the benefits of extending the tree structure to directed acyclic graphs, which may inspire other novel tree-structured models in the future.

**Limitations**   The proposed TnT decision graph is scalable to large datasets and has a linear time complexity to the number of nodes in the graph. However, the training of TnT is considerably slower than CART. The current TnT algorithm is implemented in Python. It takes about 5 minutes to construct a TnT decision graph with $\sim$1k splits on the MNIST classification task (train/test accuracy: 95.9%/90.4%). Training a CART with the same number of splits requires 12.6 seconds (train/test accuracy: 93.6%/88.3%). TnT has a natural disadvantage in terms of training time since each node is optimized multiple times (in this work $N_1 * N_2 = 10$), similar to other non-greedy tree optimization algorithms (e.g., 1-4 minutes for TAO [6]). The Python implementation may also contribute to the slow training, and we expect that the training time would significantly improve with an implementation in C. The longer training time prevents us from constructing large TnT-Adaboost ensembles that follow a sequential training process, where the training time increases linearly to the number of base estimators. We also observe that TnT decision graphs have longer decision paths compared to CART (Figure 3(a)), which may raise a concern on increased inference time. Since TnT uses a pruning factor ($C$) to control the model complexity, we can not precisely control the number of nodes as in CART. In the experiments, we searched over a range of $C$ values to meet the model complexity constraints.

**Parallel implementation**   Algorithm 2 presents a sequential algorithm to construct TnT decision graphs by visiting the nodes in the breadth-first order. However, it is also possible to concurrently grow micro decision trees inside multiple nodes, which could lead to a parallel implementation of TnT. Specifically, only those nodes in the graph that are non-descendant of each other can be optimized in parallel. Parallel optimization is not applicable to the nodes on the same decision path, since the parent node optimization may alter the samples visiting the child node. The parallel optimization of non-descendant nodes follows the separability condition of TAO [1, 6]. The separability condition also holds for the proposed TnT decision graph, enabling a parallel implementation.

## 7   Conclusion

In this paper, we propose the Tree in Tree decision graph as an effective alternative to the widely used decision trees. Starting from a single leaf node, the TnT algorithm recursively grows decision trees to construct decision graphs, extending the tree structure to a more generic directed acyclic graph. We show that the TnT decision graph outperforms the axis-aligned decision trees on a number of benchmark datasets. We also incorporate TnT decision graphs into popular ensemble methods such as bagging and AdaBoost, and show that in practice, the ensembles could also benefit from using TnTs as base estimators. Our results suggest the use of decision graphs rather than conventional decision trees to achieve superior classification performance, which may potentially inspire other novel tree-structured models in the future.

## Acknowledgments and Disclosure of Funding

We thank Dr Masoud Farivar from Google for his valuable feedback and comments on this manuscript. This work was partially supported by a Google faculty research award in machine learning.

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
