# Supplementary Materials for "Tree in Tree: from Decision Trees to Decision Graphs"

## A  Pseudocode to fine-tune TnT decision graph

We propoed the TnT algorithm to construct a decision graph from scratch. The TnT decision graph can be further fine-tuned using alternating optimization [1]. As opposed to TnT, TnT fine-tuning requires a predefined graph structure as input. A comparison between TnT and TnT(fine-tuned) is presented in Fig. 4, where TnT(fine-tuned) slightly improves both train and test accuracy. Algorithm A.1 shows the pseudocode to fine-tune TnT. Similar to Algorithm 2 in the main text, the TnT fine-tune algorithm also computes the subset $\mathbb{X}_{subset}, \mathbb{Y}_{subset}$ at each node. The hyperparameter $N$ is the number of rounds for TnT fine-tune and we fix $N = 5$ for all experiments in Fig. 4.

---

**Algorithm A.1:** Tree in Tree fine-tune

---

**Data:** Training set $\mathcal{X}, \mathcal{Y}$
**Input:** TnT decision graph $G$
**Result:** TnT decision graph $G'$ fine-tuned on $\mathcal{X}, \mathcal{Y}$

1  $\{infer(n, \mathcal{X})$ denotes the forward inference of data $\mathcal{X}$ starting from node $n\}$;
2  $\{$Nodes are visited in the breadth-first order$\}$;
3  **for** $i \leftarrow 1$ **to** $N$ **do**
4      **for** *each node* $(n_i) \in G$ **do**
5          Samples that visit $n_i$: $\mathcal{X}_i, \mathcal{Y}_i \subset \mathcal{X}, \mathcal{Y}$;
6          **if** $n_i$ *is an internal node* **then**
7              $\mathcal{Y}_{i,left} \leftarrow infer(n_i.left\_child, \mathcal{X}_i)$;
8              $\mathcal{Y}_{i,right} \leftarrow infer(n_i.right\_child, \mathcal{X}_i)$;
9              $index\_left \leftarrow (\mathcal{Y}_i = \mathcal{Y}_{i,left}$ **and** $\mathcal{Y}_i \neq \mathcal{Y}_{i,right})$ ;
10             $index\_right \leftarrow (\mathcal{Y}_i = \mathcal{Y}_{i,right}$ **and** $\mathcal{Y}_i \neq \mathcal{Y}_{i,left})$ ;
11             $\mathcal{X}_{subset}, \mathcal{Y}_{subset} \leftarrow$ copy samples from $\mathcal{X}_i, \mathcal{Y}_i$ at $index\_left$ **or** $index\_right$;
12             $\mathcal{Y}_{subset}[index\_left] \leftarrow 0, \mathcal{Y}_{subset}[index\_right] \leftarrow 1$;
13             Update the split function of $n_i$ based on $\mathcal{X}_{subset}, \mathcal{Y}_{subset}$ ;
14         **else if** $n_i$ *is a leaf node* **then**
15             $\mathcal{X}_{subset} \leftarrow \mathcal{X}_i, \mathcal{Y}_{subset} \leftarrow \mathcal{Y}_i$;
16             Label the leaf $n_i$ as the dominant class of $\mathcal{Y}_{subset}$;

---

## B  Hyperparameters of TnT

The TnT algorithm has three hyperparameters. $N_1$ is the number of merging phases where we merge micro trees into the graph. $N_2$ is the number of rounds to grow and optimize micro trees. The choice of $N_1$ and $N_2$ reflects the trade-off between training time and classification performance. We empirically set $N_1 = 2, N_2 = 5$ for all experiments in this work. $C$ is the cost complexity pruning coefficient to tune the complexity of TnT decision graphs [2, 3]. With greater $C$, TnT tends to have fewer splits. For example, Fig. 5 in the main text visualizes various model complexities with 20, 129 and 1046 splits, which is achieved with $C = 1e - 2$, $C = 1e - 3$ and $C = 1e - 4$, respectively.

35th Conference on Neural Information Processing Systems (NeurIPS 2021).

Figure 4 in the main text plots the classification performance as a function of model complexity. We tuned $C$ to change the number of splits. For each dataset, we sampled 30 values of $C$ which are equally spaced on a log scale. The maximum and minimum values of $C$ are summarized in Table B.1.

Table B.1: The maximum and minimum values of $C$ on different datasets.

| Dataset | MNIST | Connect-4 | Letter | Optical recognition | Pendigits | Protein | SenseIT | USPS |
|---|---|---|---|---|---|---|---|---|
| $C_{min}$ | 1e-4 | 6e-5 | 5e-5 | 3e-4 | 5e-4 | 8e-4 | 3e-4 | 8e-4 |
| $C_{max}$ | 5e-2 | 1e-2 | 2e-2 | 6e-2 | 1e-1 | 1e-2 | 1e-2 | 3e-2 |

In addition to using TnTs as stand-alone classifiers, we combine TnT decision graphs with ensemble methods and present TnT-bagging and TnT-AdaBoost. Additional hyperparameters are introduced to TnT-bagging and TnT-AdaBoost by the ensemble methods. In this work, we tuned the number of base estimators and the total number of splits to change the ensemble complexity. For the bagging ensemble, we randomly draw samples from the training set with replacement to train each base estimator. We set *max_samples* to 1.0 and *bootstrap_features=False* for both Random Forest and TnT-bagging. For the AdaBoost ensemble, we used the *"SAMME"* algorithm with a learning rate of 1.0 to build both AdaBoost and TnT-AdaBoost. Both ensemble methods were implemented using the scikit-learn library in Python [4].

## C    Comparison of TnT and DT ensembles

Table C.1 is similar to Table 2 in the main text but includes additional datasets. A summary on model comparison is given in the last two rows. The results show that both bagging and AdaBoost ensembles benefit from using the TnT as a base estimator.

## References

[1] Miguel A Carreira-Perpinán and Pooya Tavallali. Alternating optimization of decision trees, with application to learning sparse oblique trees. *Advances in Neural Information Processing Systems*, 31:1211–1221, 2018.

[2] Jeffrey P Bradford, Clayton Kunz, Ron Kohavi, Cliff Brunk, and Carla E Brodley. Pruning decision trees with misclassification costs. In *European Conference on Machine Learning*, pages 131–136. Springer, 1998.

[3] B Ravi Kiran and Jean Serra. Cost-complexity pruning of random forests. In *International Symposium on Mathematical Morphology and Its Applications to Signal and Image Processing*, pages 222–232. Springer, 2017.

[4] F. Pedregosa, G. Varoquaux, A. Gramfort, V. Michel, B. Thirion, O. Grisel, M. Blondel, P. Prettenhofer, R. Weiss, V. Dubourg, J. Vanderplas, A. Passos, D. Cournapeau, M. Brucher, M. Perrot, and E. Duchesnay. Scikit-learn: Machine learning in Python. *Journal of Machine Learning Research*, 12:2825–2830, 2011.


Table C.1: Comparison of TnT ensembles with random forest and AdaBoost. Mean train and test accuracy (± standard deviation) is calculated across 5 independent trials. We tune the ensemble size (#E, the number of base estimators) and split count (#S) to change the complexity of the ensembles. Dataset statistics are given in the format: Dataset name (# Train/Test samples * # Features, # Classes).

| model | dataset | #E | #S | train | test | dataset | #E | #S | train | test |
|---|---|---|---|---|---|---|---|---|---|---|
| TnT-bagging | MNIST (60k/10k*784, 10) | 5 | 4.8k | **97.46±0.16** | **93.65±0.24** | Connect-4 (45.3k/22.3k*126, 3) | 5 | 4.6k | **84.42±0.19** | **80.61±0.18** |
| Random Forest | | 5 | 4.8k | 96.55±0.36 | 92.31±0.57 | | 5 | 4.6k | 83.60±0.12 | 79.21±0.19 |
| TnT-AdaBoost | | 5 | 640 | 90.26 | 88.38 | | 5 | 450 | **77.75±0.16** | **77.39±0.19** |
| AdaBoost | | 5 | 640 | 89.75 | **88.61** | | 5 | 450 | 77.28 | 76.74 |
| TnT-bagging | | 10 | 9.6k | **98.28±0.06** | **94.92±0.20** | | 10 | 9.2k | **85.11±0.05** | **81.44±0.14** |
| Random Forest | | 10 | 9.6k | 97.44±0.18 | 93.64±0.38 | | 10 | 9.2k | 84.21±0.12 | 79.85±0.20 |
| TnT-AdaBoost | | 10 | 1.4k | **95.09±0.09** | **92.36±0.13** | | 10 | 940 | **80.10±0.23** | **78.94±0.29** |
| AdaBoost | | 10 | 1.4k | 94.28 | 91.49 | | 10 | 940 | 79.69 | 78.37 |
| TnT-bagging | | 20 | 19.2k | **98.64±0.06** | **95.57±0.14** | | 20 | 18.3k | **85.66±0.12** | **81.93±0.13** |
| Random Forest | | 20 | 19.2k | 97.90±0.12 | 94.36±0.19 | | 20 | 18.3k | 84.57±0.08 | 80.39±0.09 |
| TnT-AdaBoost | | 20 | 2.9k | **98.03±0.11** | **94.49±0.21** | | 20 | 1.8k | 82.46±0.41 | 80.53±0.50 |
| AdaBoost | | 20 | 2.9k | 97.70 | 94.04 | | 20 | 1.8k | **82.77** | **81.14** |
| TnT-bagging | | 100 | 111k | 99.09±0.03 | **96.11±0.09** | | 100 | 143k | 88.44±0.07 | **82.84±0.02** |
| Random Forest | | 100 | 292k | **100** | 95.72±0.17 | | 100 | 718k | **100** | 82.33±0.10 |
| TnT-bagging | Letter (13.4k/6.6k*16, 26) | 5 | 5.3k | 98.08±0.12 | **89.97±0.37** | Optical recognition (3.8k/1.8k*64, 10) | 5 | 890 | **99.48** | 90.45±1.24 |
| Random Forest | | 5 | 5.3k | **98.16±0.11** | 89.93±0.25 | | 5 | 890 | 99.38±0.11 | **90.46±0.91** |
| TnT-AdaBoost | | 5 | 440 | **74.51±0.83** | **73.58±0.63** | | 5 | 200 | **96.74±0.29** | **88.31±0.61** |
| AdaBoost | | 5 | 440 | 73.40 | 71.38 | | 5 | 200 | 96.73 | 87.87 |
| TnT-bagging | | 10 | 10.6k | **99.16±0.10** | **92.35±0.15** | | 10 | 1.8k | **99.83** | **92.41±0.51** |
| Random Forest | | 10 | 10.6k | 99.10±0.08 | 91.92±0.33 | | 10 | 1.8k | 99.79±0.10 | 92.23±0.37 |
| TnT-AdaBoost | | 10 | 900 | **82.90±0.38** | **80.02±0.33** | | 10 | 420 | **99.81±0.06** | 92.87±0.65 |
| AdaBoost | | 10 | 900 | 81.10 | 78.09 | | 10 | 420 | 99.58 | **92.92±0.02** |
| TnT-bagging | | 20 | 21.3k | **99.57±0.04** | **93.35±0.19** | | 20 | 3.6k | **99.91** | **92.93±0.41** |
| Random Forest | | 20 | 21.3k | 99.33±0.03 | 92.85±0.21 | | 20 | 3.6k | 99.84±0.06 | 92.78±0.23 |
| TnT-AdaBoost | | 20 | 1.8k | **90.89±0.67** | **85.33±0.56** | | 20 | 820 | **99.99±0.01** | **94.52±0.55** |
| AdaBoost | | 20 | 1.8k | 89.84 | 84.75 | | 20 | 820 | 99.97 | 94.50±0.02 |
| TnT-bagging | | 100 | 108k | 99.78±0.02 | **94.37±0.03** | | 100 | 18k | 99.93±0.03 | **93.62±0.17** |
| Random Forest | | 100 | 136k | **100** | 94.29±0.07 | | 100 | 19k | **100** | 93.37±0.24 |
| TnT-bagging | Pendigits (7.5k/3.5k*16, 10) | 5 | 570 | **99.32±0.11** | **94.12±0.27** | Protein (11.9k/5.9k*357, 3) | 5 | 1.4k | **77.05±0.58** | 59.59±0.62 |
| Random Forest | | 5 | 570 | 98.86±0.12 | 92.77±0.41 | | 5 | 1.4k | 77.30±0.53 | **59.67±0.33** |
| TnT-AdaBoost | | 5 | 200 | **98.53±0.14** | **93.24±0.62** | | 5 | 140 | **63.99** | **59.29** |
| AdaBoost | | 5 | 200 | 97.66 | 92.31 | | 5 | 140 | 62.43 | 58.45 |
| TnT-bagging | | 10 | 1.1k | **99.54±0.10** | **94.81±0.19** | | 10 | 2.7k | 80.87±0.40 | **62.75±0.25** |
| Random Forest | | 10 | 1.1k | 99.01±0.13 | 93.47±0.33 | | 10 | 2.7k | **80.88±0.28** | 62.60±0.33 |
| TnT-AdaBoost | | 10 | 410 | 99.52±0.22 | **94.83±0.21** | | 10 | 270 | **67.47** | **61.16** |
| AdaBoost | | 10 | 410 | **99.65** | 94.75±0.02 | | 10 | 270 | 66.76 | 60.92 |
| TnT-bagging | | 20 | 2.3k | **99.61±0.05** | **95.48±0.16** | | 20 | 5.4k | **83.20±0.47** | **64.44±0.44** |
| Random Forest | | 20 | 2.3k | 99.16±0.10 | 93.71±0.24 | | 20 | 5.4k | 82.82±0.24 | 64.06±0.20 |
| TnT-AdaBoost | | 20 | 820 | 100 | 96.35±0.30 | | 20 | 580 | **73.15** | 62.92 |
| AdaBoost | | 20 | 820 | 100 | **96.63** | | 20 | 580 | 72.03 | **64.03** |
| TnT-bagging | | 100 | 11k | 99.69±0.04 | **95.69±0.16** | | 100 | 0.3k | 86.71±0.21 | **66.63±0.30** |
| Random Forest | | 100 | 20k | **100** | 95.31±0.22 | | 100 | 1.5k | **100** | 66.34±0.09 |
| TnT-bagging | SenseIT (78.8k/19.7k*100, 3) | 5 | 910 | **83.92±0.12** | **82.27±0.12** | USPS (7.3k/2k*256, 2) | 5 | 540 | **98.44±0.13** | **91.29±0.34** |
| Random Forest | | 5 | 910 | 83.06±0.18 | 80.95±0.31 | | 5 | 540 | 97.27±0.16 | 90.06±0.39 |
| TnT-AdaBoost | | 5 | 110 | **77.98** | **77.47** | | 5 | 160 | **99.07** | **91.73** |
| AdaBoost | | 5 | 110 | 77.83 | 77.03 | | 5 | 160 | 97.63 | 90.53 |
| TnT-bagging | | 10 | 1.8k | **84.52±0.08** | **82.87±0.20** | | 10 | 1.1k | **98.75±0.06** | **91.90±0.16** |
| Random Forest | | 10 | 1.8k | 83.48±0.18 | 81.41±0.22 | | 10 | 1.1k | 97.85±0.19 | 90.53±0.26 |
| TnT-AdaBoost | | 10 | 170 | **79.06** | **78.46** | | 10 | 350 | **100** | **92.83** |
| AdaBoost | | 10 | 170 | 78.82 | 78.21 | | 10 | 350 | 99.95 | 92.50±0.40 |
| TnT-bagging | | 20 | 3.6k | **84.88±0.03** | **83.19±0.13** | | 20 | 2.2k | **99.20±0.09** | **92.72±0.39** |
| Random Forest | | 20 | 3.6k | 83.77±0.15 | 81.64±0.19 | | 20 | 2.2k | 98.16±0.04 | 91.29±0.42 |
| TnT-AdaBoost | | 20 | 280 | **80.00** | 79.18 | | 20 | 740 | 100 | **94.37** |
| AdaBoost | | 20 | 280 | 79.96 | **79.19** | | 20 | 740 | 100 | 94.03±0.25 |
| TnT-bagging | | 100 | 116k | 90.92±0.02 | **84.09±0.09** | | 100 | 11k | 99.29±0.05 | **93.18±0.28** |
| Random Forest | | 100 | 590k | **99.98** | 83.83±0.11 | | 100 | 24k | **100** | 92.67±0.28 |

| Summary | **TnT-bagging wins** | **test accuracy: 31** | Random Forest wins | test accuracy: 1 |
|---|---|---|---|---|
| | **TnT-AdaBoost wins** | **test accuracy: 18** | AdaBoost wins | test accuracy: 6 |

(a) Did you state the full set of assumptions of all theoretical results? [N/A]

(b) Did you include complete proofs of all theoretical results? [N/A]

3. If you ran experiments...

    (a) Did you include the code, data, and instructions needed to reproduce the main experimental results (either in the supplemental material or as a URL)? [Yes] We include a code in the supplementary material. Datasets are publicly available on UCI repository and LIBSVM Data.

    (b) Did you specify all the training details (e.g., data splits, hyperparameters, how they were chosen)? [Yes] Data splits are discussed in Section 4. Choice of hyperparameters is discussed in the supplementary materials Section B.

    (c) Did you report error bars (e.g., with respect to the random seed after running experiments multiple times)? [Yes] Figure 4 and Table 1 report standard deviations across different trials.

    (d) Did you include the total amount of compute and the type of resources used (e.g., type of GPUs, internal cluster, or cloud provider)? [Yes] Platform and training time are reported in Section 3, Time complexity.

4. If you are using existing assets (e.g., code, data, models) or curating/releasing new assets...

    (a) If your work uses existing assets, did you cite the creators? [Yes] See reference [28], [29]

    (b) Did you mention the license of the assets? [Yes] The scikit-learn library is under the 3-Clause BSD license. Some datasets (e.g., MNIST) are under Creative Commons Attribution-Share Alike 3.0 license.

    (c) Did you include any new assets either in the supplemental material or as a URL? [Yes]

    (d) Did you discuss whether and how consent was obtained from people whose data you're using/curating? [N/A] All datasets are publicly available on UCI repository and LIBSVM Data.

    (e) Did you discuss whether the data you are using/curating contains personally identifiable information or offensive content? [N/A]

5. If you used crowdsourcing or conducted research with human subjects...

    (a) Did you include the full text of instructions given to participants and screenshots, if applicable? [N/A]

    (b) Did you describe any potential participant risks, with links to Institutional Review Board (IRB) approvals, if applicable? [N/A]

    (c) Did you include the estimated hourly wage paid to participants and the total amount spent on participant compensation? [N/A]