# OpenReview forum: "Tree in Tree: from Decision Trees to Decision Graphs"
_NeurIPS.cc/2021/Conference — NeurIPS 2021 Poster_

### Official Review · Reviewer_nBwC · 2021-07-14

**Rating:** 6
**Confidence:** 4

**Summary:**

The authors present a new algorithm for large decision graphs. Tree in a tree is a scalable method that fits new decision trees in place of the internal/leaf nodes and constructs a directed acyclic graph. The authors show that their method outperforms other learning models in different datasets with different configurations.

**Ethical Concerns:**

-

**Limitations And Societal Impact:**

Yes.

**Main Review:**

* Positive points:
- The paper is well written and well structured. The authors are clear and objective in their communication.
- The experimental procedure sounds robust, with various datasets and classification methods (decision trees/graphs).
- The time complexity sounds good, even though the authors state very clearly that CART and TAO would be more efficient.

* Negative Points:
- Since the authors are using accuracy to quantify the power of their method, I believe more classification methods should be used in their study.  If the authors were accessing the readability of the final graph, it would be understandable to only use other decision trees and graphs as the benchmark. However,  accuracy-wise, other classification methods (e.g., neural networks and kernel machines) should be used as well.
- Considering that decision trees are sometimes used for their explainability, I would have liked to see each dataset's result. Figure 5 is not readable or understandable.
- I do not believe that the limitations written by the authors are limitations at all. A trade-off is clear for me in terms of accuracy versus training time. I would have like to see more limitations on this work or a more robust argumentation on why this limitation is essential for TnT.

**Time Spent Reviewing:**

8 hours

---

> ### Author Response · Authors · 2021-08-10
> **Response to Reviewer nBwC**
>
> We thank the reviewer for the comments. We provide the following response to the reviewer’s concern:
> * As the reviewer pointed out, there exist other models which are comparable with TnT in terms of classification accuracy. However, models such as neural networks and kernel machines do not use axis-aligned split boundaries as TnT. It should be noted that the core idea of TnT (i.e., growing micro trees inside nodes) is generic and compatible with more complex split functions (discussed in the broader impact section). Moreover, the model complexity of neural networks and kernel machines can not be quantified using the number of splits. For an apple-to-apple comparison, we only included interpretable models that use axis-aligned split functions in the internal nodes (e.g., CART).
> * We will release the TnT visualization code along with the camera-ready version of the paper. Users can easily visualize models trained on their target datasets. Since MNIST has many features and classes, we agree that interpretation is hard. We will provide more visualization examples, particularly on small datasets for better readability. We will also increase the resolution of figures and optimize node layout. The root node will be marked out.
> * Compared to CART and TAO, the proposed TnT has longer training time. We agree that this is more of a tradeoff than a limitation. The training time of TnT is acceptable as a stand-alone classifier. However, ensemble methods usually include several TnTs. For example, TnT-Adaboost follows a sequential training process where the total training time increases linearly to the number of base estimators. Therefore, longer training time could limit the number of TnTs in the ensemble. Moreover, since TnT uses a pruning factor $C$ to control the model complexity, we can not precisely control the number of nodes as in CART. In the experiments, we searched over a range of $C$ to meet model complexity constraints. We will add this point to the limitation section.

---

> > ### Comment · Reviewer_nBwC · 2021-09-02
> > **Response do authors**
> >
> > I thank the authors for their response. They have addressed my concerns satisfactorily.

---

### Official Review · Reviewer_Vnre · 2021-07-14

**Rating:** 6
**Confidence:** 4

**Summary:**

The paper proposes an algorithm to build decision acyclic graphs with properties similar to those of decision trees: each node has a split condition to route instances and terminal nodes of the graph (i.e. those without children) provide the classification labels.  This model is trained by creating embedded decision trees inside decision the graph nodes iteratively. The algorithm starts with a single leave node in which, a small decision tree is build to route the examples to left/right. Then the decision tree internal decision tree of the node is flattened to create a graph. The acyclic graph structure appears since all left/right decisions of the embedded tree are routed to the same left/right graph node. Then, for each node the same process is applied iteratively.

The method is used as the base classifier of bagging and Adaboost ensembles and compared on 8 datasets against standard decision trees with RF and Adaboost using the same over all complexity (complexity measured as number of splits).

**Main Review:**

The paper is well written and quite clear, although some aspects could be further clarified (see below).

I have some issues/doubts bout the paper:
* The description of algorithm 1 is partial and a bit unclear. For instance, N1 and N2 are not described when the algorithm is introduced. I believe that N1 is the number of times the DT in the nodes of the graph are flattened. N2 remain unclear, why does the algorithm goes N2 times over all nodes of the graph to build a DT at each node? Also in line 2, t subscript appears but it is not defined before, Should be 0 instead of t? In addition, infer function is only explained with a comment in the algorithm. I suppose it is related to eq (6).
* One unclear aspect to me is how he complexity of the models are gauged for the experiments. I understand that there is a C hyper-param that could be used to limit the complexity of the mini decision trees but it is unclear to me how to limit the over all complexity of the decision graph. For intance, are the curves in Fig3(c,d) obtained modifying C or do they use a fixed C? If so, which one? if not, how the C is modified? Idem for Fig4. Also in Fig4, how are the complexity of CART controlled? Using C? max_depth? other? A curious aspect is that for most plots, there is o over-fitting with the increment of the complexity of trees and graphs.
* The article says "With 1000 splits, the average decision depth of the best-first CART is 12.3,whereas the TnT decision graph has a mean depth of 27.3." Is it a good thing that the paths are longer? How about the performance of DT and DG at equal path lengths?
* The comparison of TnT +bagging/adaboost with standard RF and adaboost is done in the context of equal complexity in terms of number of base models and overall number of splits. Again, the process to achieve this equal complexity is unclear. Is the C set for TnT ensembles and then using the obtained complexity, the standard ensembles are created with different max_depth (?) until they reach the same complexity? Any way I believe the comparison has to be done also without constrains for both methods. Using a maximum number of base DT of 20 (#E) is clearly suboptimal for RF and adaboost. Also using a limiting number of split for RF is suboptimal. The results of a fully trained RF (#E=100-200, and no splits limit) and adaboost should also be reported.

Minor aspects:
* Sometimes the paper is a bit repetitive (e.g. lines 164-171).
* Figure 2 plots (N 1 = 2, N 2 = 5) in c) and d) with different colour. it would be clearer to use the same colour.
* Fig 5: It could be interesting to mark the root node in the plots. Also plot b) could be unentanled a bit to better visualize.


**Time Spent Reviewing:**

4

---

> ### Author Response · Authors · 2021-08-10
> **Response to Reviewer Vnre**
>
> We thank the reviewer for the comments. We made the following clarifications in response to these comments.
>
> ## Algorithm clarification
> We included a discussion on hyperparameters in Section B of the supplementary materials. To better clarify $N_1$ and $N_2$, we will move the discussion to the Methods section. Overall, the TnT training algorithm follows an alternating optimization procedure. $N_1$ is the number of merging phases where we merge micro trees into the graph (i.e., micro tree flattened). $N_2$ is the number of rounds to grow and optimize micro trees. $N_2$ is similar to the number of iterations in the tree alternating optimization algorithm [1, 5]. In Algorithm 2, we optimize the micro tree at one node ($n_i$) for each step while keeping other nodes fixed. After the whole graph is updated, the trained micro tree at $n_i$ may be suboptimal. Therefore, we revisit $n_i$ and repeat the micro tree optimization process for $N_2$ iterations for the tree alternating optimization to converge. The effect of tuning $N_1$ or $N_2$ is shown in Figure 3(c, d).
>
> In Algorithm 2 line 2, the t subscript indicates an arbitrary node index $t \in$ \{1,...,graph size\}. The $infer()$ function indicates the inference process of TnT decision graphs, which is identical to trees. The inference process starts from the node $n_t$ which is an input argument of $infer()$. Equation (6) is the training objective of micro trees in leaf nodes, which takes the prediction as input. We will clarify these points and improve the notations in the final manuscript.
>
> ## The effect of hyperparameter $C$ in TnT
> As the reviewer pointed out, we used the hyperparameter $C$ to control the complexity of micro trees. However, since the micro trees will be merged (i.e., flattened) into the TnT graph, the complexity of TnT is determined by the size of the micro trees. Specifically, the curves in Figure 3(c, d) and Figure 4 are plotted by modifying $C$. Each point on the curve corresponds to a value of $C$. We sampled 30 values of $C$ which are equally spaced on a log scale. In Figure 3(c, d), the range of $C$ is from 1e-4 to 5e-2. The maximum and minimum values of $C$ are summarized in Table B.1 of the supplementary material for each dataset listed in Figure 4.
>
> ## CART complexity control
> For CART, we used a best-first growing scheme and tuned the maximum number of leaf nodes (max_leaf_nodes) to change the model complexity. In Figure 4, we plotted the classification performance of algorithms at various model complexity levels for a fair, apple-to-apple comparison. To further address the concern on overfitting, we added a new experiment that compares models at their optimal complexity. The optimal complexity is determined by 3-fold cross-validation on the training set. Therefore, we will add the optimal complexity to Figure 4 in order to better identify overfitting on the plots. We will increase the maximum number of splits on some plots to make overfitting more visible. Overall, TnT achieved better accuracy with reduced model complexity compared to CART. We provide the detailed comparison on the first 2 datasets as an example (averaged across 5 independent trials). Comparisons on more datasets will be made available in our GitHub repository.
>
> Dataset | Model | Train Accuracy| Test Accuracy | # Splits
> --- | --- | --- | --- | ---
> MNIST | CART(stand-alone) | 94.04±0.22 | 88.59±0.14 | 1.1k
> MNIST | TnT(stand-alone) | 94.05±0.56 | 90.87±0.31 | 600
> Connect-4 | CART(stand-alone) | 81.60 | 77.23±0.01 | 931
> Connect-4 | TnT(stand-alone) | 83.36±1.58 | 78.85±0.46 | 864
>
> ## Depth of TnT graph
> In this work, we experimentally verified that TnT is “deeper” with longer decision paths. In the limitation section (lines 315-316), we mentioned that longer decision paths may raise a concern about increased inference time of TnT. However, on the bright side, long decision paths allow small TnT models to represent a “deep” model and achieve strong predictive power, which may explain the superior performance of TnT under model complexity constraints. To compare DT and DG at equal path length, we recursively grow a CART tree until all leaf nodes are pure. Therefore, the resulting DT has the largest number of nodes (3937) and the longest decision path (23.3, still shorter than TnT 27.3). With a similar path length, CART is heavily overfitted and TnT shows superior performance in terms of both model complexity and test accuracy. The detailed comparison is listed below:
>
> MNIST | Train Accuracy| Test Accuracy | Path Length | # Splits
> --- | --- | --- | --- | ---
> CART | 100 | 87.67 | 23.3 | 3937
> TnT | 96.04 | 90.56 | 27.3 | 1046
>
> ## Experimental comparison of TnT/CART ensembles
> For the ensemble experiments, we first set various model complexity constraints for both TnT and CART ensembles. We tuned $C$ for TnT and max_leaf_nodes for CART to fit both models into various complexity constraints (e.g., number of base estimators and splits). Therefore, we compared the performance of TnT-bagging/Adaboost against RF/Adaboost at the same complexity level.
>
> The reviewer raised a concern that the performance of RF may be suboptimal due to the limit on #E (number of base estimators) and #S (number of splits). To address this concern, we added an additional experiment to relax the complexity constraints. As suggested by the reviewer, we used 100 trees for ensembles. The limit on #S is completely removed for RF and trees are allowed to grow as large as possible. Overall, the comparison result is consistent with Table 1, where TnT-bagging achieved better accuracy with reduced model complexity compared to RF. Since Adaboost is not compatible with parallel training, building a large Adaboost ensemble takes a long time. Adaboost comparisons will be updated later. We attach the detailed comparison on MNIST as an example (averaged across 5 independent trials). Comparisons on more datasets and with more base estimators will be made available in our GitHub repository.
>
> MNIST | Train Accuracy| Test Accuracy | # Estimators | # Splits
> --- | --- | --- | --- | ---
> Random Forest | 100 | 95.72±0.17 | 100 | 292k
> TnT-Bagging | 99.09±0.03 | 96.11±0.09 | 100 | 111k
>
> ## Minor comments
> We will proofread the manuscript to remove repetitive statements. Figures 2 and 5 will also be replotted for improved readability.

---

### Official Review · Reviewer_gGfV · 2021-07-16

**Rating:** 7
**Confidence:** 4

**Summary:**

Inspired from Network in Network the paper proposes an extension of decision trees that replaces internal and leaf nodes with micro decision trees, thus extending decision trees to decision graphs.


**Limitations And Societal Impact:**

The main limitation regards the time complexity of the proposed approach.

**Main Review:**

In particular the authors propose a learning algorithm for decision graphs adopting a simple and effective iterative algorithm.

It should be interesting to explain why a decision graph has an accuracy greater than that of decision tree. Since there are shared decision paths, it seems that the decision power of a graph is lower than that of a tree. Probably it is due to deeper models involving more splits on the path to the leaf. A discussion should be interesting.

The learning approach presented in section 3 is clear and well presented.

Finally, the experimental evaluation proves that the TnT decision graph outperforms the axis-aligned decision trees on a number of benchmark datasets.

**Time Spent Reviewing:**

3

---

> ### Author Response · Authors · 2021-08-10
> **Response to Reviewer gGfV**
>
> We thank the reviewer for the comments. The advantages of TnT decision graphs over conventional decision trees can be discussed from the following perspectives:
> * Since nodes are shared, the model size of TnT is smaller than trees. Therefore, under the same model complexity constraint, TnT can represent a more complex and accurate model than the tree counterpart.
> * The node-sharing mechanism can be considered as a regularization, which can effectively reduce variance.
> * Our experimental results show that TnTs grow much deeper than standard trees. We conducted an additional experiment to show this fact quantitatively. On MNIST, TnT has a mean depth of 27.3 with only 1k splits. However, if we fully grow a CART tree such that all leaf nodes are pure, the depth of CART is 23.3 with 3.9k nodes. This result shows that TnT can grow deeper with fewer nodes, which could account for its superior performance.

---

### Official Review · Reviewer_GefL · 2021-07-19

**Rating:** 6
**Confidence:** 3

**Summary:**

This paper proposes an algorithm to build decision graphs. The idea of this algorithm is to recursively replace internal nodes as well as leaf nodes in a decision tree by a small decision tree. In the case of internal nodes, this decision tree is grown to optimally direct the examples falling into that node to the left or right successor of the replaced node. In the case of leaf nodes, the small tree is grown to predict the target. Experiments compare the approach with standard CART tree and ensembles.


**Limitations And Societal Impact:**

Yes.

**Main Review:**

The idea of building decision graphs is not new but it is interesting. The way the authors motivate them in the paper is however not very convincing. They draw a parallel between TnT and oblique trees, which suggest that TnT are more expressive that standard trees and that's the reason they are expected to be more accurate. This is actually not the case, since any decision graph can indeed be represented by a (more complex) decision tree. In my opinion, the main interest of graphs instead of trees is the reduced size of the model and the potential reduction of variance (because using a graph allows to avoid the sample size reduction effect inherent to tree partitioning).

The proposed algorithm to grow graphs is original to the best of my knowledge. The related work section is however incomplete. There are a number of other decision graph training algorithms, such as the following two (and several other references therein):
* Zighed D.A. (2007) Induction Graphs for Data Mining. In: Brito P., Cucumel G., Bertrand P., de Carvalho F. (eds) Selected Contributions in Data Analysis and Classification. Studies in Classification, Data Analysis, and Knowledge Organization. Springer, Berlin, Heidelberg.
* Jamie Shotton, Sebastian Nowozin, Toby Sharp, John Winn, Pushmeet Kohli, and Antonio Criminisi. 2013. Decision jungles: compact and rich models for classification. In Proceedings of the 26th International Conference on Neural Information Processing Systems - Volume 1 (NIPS'13).

I think the proposed method should be discussed and compared more thoroughly with this literature.

In the end, I think I managed to understand the main idea of the algorithm but the description could have been clearer. What confused me the most was how internal splits are expanded into micro-trees, i.e., the way labels are constructed to train these micro-trees (as explained in 3.1). Part of the confusion comes from the fact that some notations are not clearly explained (e.g., in (5) what is $G_{n_i\rightarrow left}$  and why does it take as its first argument a subset $X$, and not a single $x$ as $G$).

In Section 3.3, a discussion and motivation for the introduction of the parameters $N_1$ and $N_2$ in Algorithm 2 is missing and this added to my confusion (there is more discussion of these parameters in the supplementary material however). Why wait for $N_2$ steps of node expansion before merging? Actually, why not merging after each micro tree construction inside the "for each node ($n_i$)" loop? In line 17, there is a reference to $t_i$ while we are outside the loop over $n_i$. What does it mean to merge $t_i$ there? Is the indentation correct?

Also, if I understand correctly, the order in which the nodes are considered in the loop in Line 5 matters (since changing an internal node into a replacement micro-tree will change the prediction for the samples falling into that node and thus will have an impact on its ancestors). If this is correct, it's not clear from the paper which order is used in practice.

Overall, the algorithm to construct these graphs makes sense and I like the idea of trees in trees. One drawback of the approach however is the introduction of the three hyper-parameters $N_1$, $N_2$, and $C$, whose interplay is not very clear. The authors seem also to put more emphasis on setting $C$ than on setting $N_1$ and $N_2$, which are mainly fixed to arbitrary default values. I have the feeling that it would be possible to have a single hyper-parameter. For example, why setting $N_2$ to 1 is not enough?

The experiments show that there is a benefit in using graphs in terms of accuracy. Methods are compared however at fixed number of splits and I don't think it's appropriate (or at least enough). First, there is no mechanism in TnT to set precisely the number of nodes and thus, this is not a very natural comparison (in practice, this is not how the method will be used). Second, one of the main benefits of CART is that it's a non-parametric method that can adapt model complexity to the problem at hand (using pruning). I would thus compare both methods at their optimum complexity (determined by CV), in terms of accuracy, number of splits and inference time.

The same comment applies for the comparison within ensembles. The number of ensemble terms and the number of nodes is arbitrarily set to small numbers in the experiment, while in practice, RF are used with unpruned trees and ensembles of hundreds of trees are averaged. Showing that TnT ensembles are better than suboptimal forests is not a very convincing result in my opinion.

Minor remarks:
* Equation (4) is said to be an optimisation problem, which closed-form solution is (5). I don't understand notations in (5). What is optimized? Why is the sum over all examples? What is the value of $G_{n_i\rightarrow left/right}(x)$ for samples that do not reach $n_i$?
* The synthetic examples show indeed the potential benefit of using a DAG instead of a tree, if one wants to reduce the number of splits. It's not clear however whether the proposed algorithm is able to produce the graph in 2(c) for such problem given the heuristic nature of the algorithm. If not, then I think presenting this example after having presented the algorithm is a bit misleading.
* In Figure 4, are CART trees fully grown when the maximum number of splits is reached or would it be possible to grow them further? I was expecting the accur I'm not sure that the maximum number of splits.


**Time Spent Reviewing:**

4

---

> ### Author Response · Authors · 2021-08-10
> **Response to Reviewer GefL**
>
> We thank the reviewer for the thoughtful comments. We provide the following clarifications in response to these comments.
>
> ## Motivation:
> We completely agree with the reviewer that TnT is less prone to overfitting compared to trees. The node-sharing mechanism in TnT regularizes the growth of a graph, which may account for the superior test performance of TnT.  As pointed out, TnT has the advantage of reduced model size. Figure 4 shows that TnT uses fewer splits to achieve the same performance level compared to trees. Therefore, TnT is a more compact and accurate alternative to the widely-used decision trees. We will further highlight this point in our revised paper.
>
> Indeed, TnT can be represented by a larger, more complex decision tree. For resource-constrained applications where models are only allowed to have a fixed number of splits, TnT is more expressive than standard trees. In this paper, we experimentally verified that TnT is more accurate than trees under the same model complexity (Figure 4, Table 1). We connect TnT to oblique trees since both models share a similar training process, i.e., using complex split functions inside internal nodes (e.g., micro trees for TnT and linear splits for oblique trees). Recent work has verified that oblique trees can improve classification performance under a given complexity constraint (e.g., 2kB of RAM as in [2]). Here, we propose TnT with the goal of growing a small but accurate model with only axis-aligned splits.
>
> ## Missing reference:
> Previous work constructed decision graphs by merging nodes. We cited a few early papers on that track ([14, 15]) and presented the general graph construction pipeline as NDG (Algorithm 1). The suggested papers can be considered as recent variations of NDG. For example, *Zighed D.A. ’07* used node “fusion and splitting” as basic operations. *J Shotton, et al, NIPS’13* jointly optimized the split node ($n_{spilt}$) and their connections with the next-layer nodes, which could be interpreted as merging the $n_{spilt}$’s child with another node at the same layer. Our approach is different from previous methods and scalable to large graphs. To better reflect the progress of the merging-based graph construction, we will add new references as follows and expand our discussion in Section 2.
>
> * Shotton, Jamie, et al. "Decision jungles: Compact and rich models for classification." Proceedings of the 26th International Conference on Neural Information Processing Systems (2013).
> * Zighed D.A. “Induction Graphs for Data Mining.” Selected Contributions in Data Analysis and Classification. Studies in Classification, Data Analysis, and Knowledge Organization. Springer, Berlin, Heidelberg (2007).
> * Benbouzid, Djalel, Róbert Busa-Fekete, and Balázs Kégl. "Fast classification using sparse decision DAGs." arXiv preprint arXiv:1206.6387 (2012).
>
> ## Algorithm/Notation/Hyperparameter clarification
> The only difference between $G_{n_i\to left}$ and $G$ lies in the internal node $n_i$. In $G$, $n_i$ routes samples according to its split function $s_i$. In $G_{n_i\to left}$, $n_i$ always routes samples to its left child regardless of $s_i$ (lines 142-143). In the manuscript, the first argument of  $G_{n_i\to left}$ could either be one sample or a set of samples. To keep it consistent with $G$ and avoid confusion, we have updated the notation in (2) such that $G_{n_i\to left}$ takes only a single sample as input ($G_{n_i\to left}(x_{subset}; \Theta \backslash \theta_i) \neq G_{n_i\to right}(x_{subset}; \Theta \backslash \theta_i), x_{subset} \in \mathbb{X}_{subset}$). Thanks for the suggestion.
>
> A discussion on hyperparameters is included in Section B of the supplementary materials. We will move the discussion to the Methods section to better clarify $N_1$ and $N_2$ when the algorithm is introduced. Overall, the TnT training algorithm follows an alternating optimization procedure. $N_1$ is the number of merging phases where we merge micro trees into the graph. With $N_1=1$, we simply obtain a decision tree which can be considered a special form of graphs. Therefore, we need to set $N_1>1$ to demonstrate the advantage of graphs over standard trees. $N_2$ is the number of rounds to grow and optimize micro trees. If we merge the micro tree immediately after construction, it would be equivalent to $N_2=1$. In Algorithm 2, we confirm that the indentation is correct. $N_2$ is similar to the number of iterations in the tree alternating optimization algorithm as in [1, 5]. We optimize the micro tree at one node ($n_i$) for each step while keeping other nodes fixed. After the whole graph is updated, the trained micro tree at $n_i$ may become suboptimal. Therefore, we revisit $n_i$ and repeat the micro tree optimization process for $N_2$ iterations until the tree alternating optimization converges. Figure 3 (c, d) compare different values of $N_1$ and $N_2$. With $N_2=1$, the performance is slightly lower than $N_2=3$ or $N_2=5$. Higher $N_1$ and $N_2$ values generally lead to better performance at the cost of longer training time.
>
> In this paper, nodes are visited in the breadth-first order (line 318). However, previous work has reported that tree alternating optimization is robust to different node orders [5]. We will clarify the node visiting order in the Methods section.
>
> In this paper, we simply set $N_1=2,N_2=5$ and tune $C$ to change model complexity. This is because we want to make a fair comparison with standard trees where only one parameter was tuned (i.e., number of splits). In our experience, all three hyperparameters are necessary, since they control different aspects of the algorithm ($N_1$ the number of merging phases, $N_2$ the number of rounds for alternating optimization). Setting $N_2=1$ will cause non-optimality in the alternating optimization framework and setting $N_1=1$ will only grow a tree instead of a graph. Please consider our previous response for a detailed discussion.
>
> ## Appropriate comparison
> First, we agree with the reviewer that we can not precisely set the split count in TnT. We tune the pruning strength $C$ to control TnT complexity. We will add this point in the limitation section. Second, the primary goal of this paper is to compare TnT and other models under the same model complexity constraint. Figure 4 plots the performance as a function of model complexity, which allows us to compare models under fixed complexity constraints (e.g., <50, <100, or <200 splits). The comparison between TnT and CART ensembles follows the same spirit, where we trained the model under 6 different model complexity constraints. Thus, all models are regularized to fit into the same requirements.
>
> To address this comment, we added a new experiment that compares TnT and CART at their optimal complexity. We determined the optimal complexity of both TnT and CART with a 3-fold CV on the training set. The complexity requirements were completely removed and we picked the hyperparameters which achieved the highest CV accuracy. Similar to Figure 4, we repeated the experiments for 5 independent trials. Overall, TnT achieved better accuracy with reduced model complexity compared to CART. The inference time of TnT is longer than that of CART, because TnT is “deeper” than CART and our custom inference implementation (in python) is slower than the CART library with C backend. We provide below the detailed comparison on the first two datasets as an example. More comparisons are available upon request.
>
> Dataset | Model| Train Accuracy| Test Accuracy | # Splits | Inference Time(s)
> --- | --- | --- | --- | --- | ---
> MNIST | CART(stand-alone) | 94.04±0.22 | 88.59±0.14 | 1.1k | 0.01
> MNIST | TnT(stand-alone) | 94.05±0.56 | 90.87±0.31 | 600 |0.98
> Connect-4 | CART(stand-alone) | 81.60 | 77.23±0.01 | 931 | 0.05
> Connect-4 | TnT(stand-alone) | 83.36±1.58 | 78.85±0.46 | 864 | 4.89
>
>
> The reviewer raised a concern that the performance of RF may be suboptimal due to the limit on #E (number of base estimators) and #S (number of splits). To address this concern, we added an additional experiment to relax the complexity constraint. We used 100 trees in the ensembles. The limit on #S is completely removed for RF and trees are allowed to grow as large as possible. Overall, the comparison result is consistent with Table 1, where TnT ensembles achieved better accuracy with reduced model complexity compared to CART ensembles. We provide the detailed comparison on MNIST as an example (averaged across 5 independent trials). Comparisons on more datasets and with more base estimators will be made available in our GitHub repository.
>
>
> MNIST | Train Accuracy| Test Accuracy | # Estimators | # Splits
> --- | --- | --- | --- | ---
> Random Forest | 100 | 95.72±0.17 | 100 | 292k
> TnT-Bagging | 99.09±0.03 | 96.11±0.09 | 100 | 111k
>
> ## Response to minor remarks
> * In (4), we optimize the split function (replaced by micro trees in the growing phase) at node $n_i$. Equation (5) derives the micro tree objective which solves the optimization problem at $n_i$ (4). For samples that do not visit $n_i$, $G_{n_i\to left}(x)=G_{n_i\to right}(x)$. Therefore, those samples have no impact on the optimization problem at $n_i$. The summation over all samples in (4) is only to be consistent with (1). Summation over $X_{subset}$ will do the same thing. We updated (4) to specify what parameters are optimized at node $n_i$.
> * Yes. We verified that TnT can produce the exact same graph as shown in Figure 2(c). Using the demo code and TnT(N1=3, N2=1, max_leaf_nodes=3), a TnT with 4 splits and 4 leaves can perfectly fit on the synthetic data.
> * The CART trees are not fully grown in Figure 4. For example, the maximum number of splits of CART on the MNIST dataset is 3.9k. The test accuracy will not further improve as we use more splits, since models are overfitted. For the optimal complexity that maximizes CV performance, please refer to the table in the previous response.

---

> > ### Comment · Reviewer_GefL · 2021-08-29
> > **Response to the authors**
> >
> > I thank the authors for their detailed response. They answer to most of my complaints.
> >
> > Some additional comments:
> >
> > > Second, the primary goal of this paper is to compare TnT and other models under the same model complexity constraint."
> >
> > This is precisely what I criticism. It's unfair. At a fixed budget, standard trees will be obviously less good than TnT. The new results are interesting as they show that when complexity is optimized, you can obtain more accurate and simpler models with TnT. I suggest the authors to provide the same results on all datasets (with the hope that they are all going in the same direction).
> >
> > > In this paper, nodes are visited in the breadth-first order (line 318). However, previous work has reported that tree alternating optimization is robust to different node orders [5]."
> >
> > This is not a very convincing argument. Previous works did not concern building decision graphs. There is no reason to believe that the optimal order should be breadth-first here as well.
> >
> > Concerning related works, I have not doubt that the proposed approach is different from existing methods. I'm however not convinced that NDG (Algorithm 1) is representative of all existing methods, especially since it does not improve over CART. Although I like the idea of TnT, this is yet another algorithm to build a decision graph and I have no idea how it compares with all other existing methods that build similar graphs.
> >
> > That said, given the new experiments, I'm happy to raise a bit my score.

---

> > > ### Author Response · Authors · 2021-08-30
> > > **Response to Reviewer GefL**
> > >
> > > ## Additional experiments on all datasets
> > > We thank the reviewer for the feedback. We agree that the advantage of TnT over standard trees looks obvious at a fixed budget. We will include a new experiment in the final version to compare both models at their optimal complexity (determined by CV). We attach here the results on all eight datasets. TnT can consistently achieve a better test accuracy with fewer splits and the conclusion is consistent with the paper.
> > >
> > > Dataset | Model| Train Accuracy| Test Accuracy | # Splits | Inference Time(s)
> > > --- | --- | --- | --- | --- | ---
> > > MNIST | CART(stand-alone) | 94.04±0.22 | 88.59±0.14 | 1.1k | 0.01
> > > MNIST | TnT(stand-alone) | 94.05±0.56 | 90.87±0.31 | 600 |0.98
> > > Connect-4 | CART(stand-alone) | 81.60 | 77.23±0.01 | 931 | 0.05
> > > Connect-4 | TnT(stand-alone) | 83.36±1.58 | 78.85±0.46 | 864 | 4.89
> > > Letter | CART(stand-alone) | 97.41±0.35 | 86.26±0.15 | 1.3k | 0.002
> > > Letter | TnT(stand-alone) | 97.40±0.02 | 86.62±0.02 | 1.2k | 4
> > > Optical recognition | CART(stand-alone) | 98.89±0.49 | 85.56±0.46 | 193 | 0.0005
> > > Optical recognition | TnT(stand-alone) | 98.88±0.82 | 86.32±0.24 | 174 | 0.05
> > > Pendigits | CART(stand-alone) | 99.29±0.21 | 91.74±0.13 | 166 | 0.0004
> > > Pendigits | TnT(stand-alone) | 99.21±0.24 | 92.61±0.53 | 125 | 0.06
> > > SenseIT | CART(stand-alone) | 82.97 | 79.40 | 345 | 0.004
> > > SenseIT | TnT(stand-alone) | 82.35±0.35 | 80.48±0.42 | 198 | 0.6
> > > USPS | CART(stand-alone) | 96.61±0.44 | 87.35±0.15 | 109 | 0.0009
> > > USPS | TnT(stand-alone) | 93.76±2.12 | 88.76±1.36 | 31 | 0.009
> > >
> > > We also show the comparison between TnT and CART ensembles with 100 estimators and fully-grown CART on all datasets. TnT ensembles can achieve better test accuracy with fewer splits.
> > >
> > > Dataset | Model | Train Accuracy| Test Accuracy | # Estimators | # Splits
> > > --- | --- | --- | --- | --- | ---
> > > MNIST | Random Forest | 100 | 95.72±0.17 | 100 | 292k
> > > MNIST | TnT-Bagging | 99.09±0.03 | 96.11±0.09 | 100 | 111k
> > > Connect-4 | Random Forest | 100 | 82.33±0.10 | 100 | 718k
> > > Connect-4 | TnT-Bagging | 88.44±0.07 | 82.84±0.02 | 100 | 143k
> > > Letter | Random Forest | 100 | 94.29±0.07 | 100 | 136k
> > > Letter | TnT-Bagging | 99.78±0.02 | 94.37±0.03 | 100 | 108k
> > > Optical recognition | Random Forest | 100 | 93.37±0.24 | 100 | 19k
> > > Optical recognition | TnT-Bagging | 99.93±0.03 | 93.62±0.17 | 100 | 18k
> > > Pendigits | Random Forest | 100 | 95.31±0.22 | 100 | 20k
> > > Pendigits | TnT-Bagging | 99.69±0.04 | 95.69±0.16 | 100 | 11k
> > > SenseIT | Random Forest | 99.98 | 83.83±0.11 | 100 | 590k
> > > SenseIT | TnT-Bagging | 90.92±0.02 | 84.09±0.09 | 100 | 116k
> > > USPS | Random Forest | 100 | 92.67±0.28 | 100 | 24k
> > > USPS | TnT-Bagging | 99.29±0.05 | 93.18±0.28 | 100 | 11k
> > >
> > > ## Node order and related work
> > > We thank the reviewer for the positive feedback on our manuscript and agree with the comments on node order and reference to related work. We will clarify that we used the breadth-first order to visit nodes in the algorithm and experimentally analyze the effect of different node orders. Regarding related work, we will further discuss other graph-based models in the paper and include a more detailed comparison with existing methods.

---

### Decision · Program_Chairs · 2021-09-27

**Decision:**

Accept (Poster)

**Comment:**

The paper presents a way to to grow decision graphs by using decision trees as the splitting criterion of another tree. Reviewers praised the simplicity of the idea and its effectiveness on classification tasks.

During the reviewing and rebuttal phase, some criticisms regarding missing references and more crucially missing experiments with suitable comparisons were raised.
Authors provided additional convincing results in the rebuttal and managed to convince one reviewer to upgrade its weak rejection score into a weak acceptance.

The paper is accepted, subject to including the missing comparisons to the literature, and the additional baselines and experiments discussed in the rebuttal.